# A Machine Learning Method for the Quantitative Detection of Adulterated Meat Using a MOS-Based E-Nose

**DOI:** 10.3390/foods11040602

**Published:** 2022-02-20

**Authors:** Changquan Huang, Yu Gu

**Affiliations:** 1College of Information Science and Technology, Beijing University of Chemical Technology, Beijing 100029, China; huangchangquan@mail.buct.edu.cn; 2Beijing Advanced Innovation Center for Soft Matter Science and Engineering, Beijing University of Chemical Technology, Beijing 100029, China; 3Guangdong Province Key Laboratory of Petrochemical Equipment Fault Diagnosis, Guangdong University of Petrochemical Technology, Maoming 525000, China; 4Department of Chemistry, Institute of Inorganic and Analytical Chemistry, Goethe-University, Max-von-Laue-Str. 9, 60438 Frankfurt, Germany

**Keywords:** meat adulteration, electronic nose, one-dimensional convolutional neural network, random forest regressor

## Abstract

Meat adulteration is a global problem which undermines market fairness and harms people with allergies or certain religious beliefs. In this study, a novel framework in which a one-dimensional convolutional neural network (1DCNN) serves as a backbone and a random forest regressor (RFR) serves as a regressor, named 1DCNN-RFR, is proposed for the quantitative detection of beef adulterated with pork using electronic nose (E-nose) data. The 1DCNN backbone extracted a sufficient number of features from a multichannel input matrix converted from the raw E-nose data. The RFR improved the regression performance due to its strong prediction ability. The effectiveness of the 1DCNN-RFR framework was verified by comparing it with four other models (support vector regression model (SVR), RFR, backpropagation neural network (BPNN), and 1DCNN). The proposed 1DCNN-RFR framework performed best in the quantitative detection of beef adulterated with pork. This study indicated that the proposed 1DCNN-RFR framework could be used as an effective tool for the quantitative detection of meat adulteration.

## 1. Introduction

Meat is one of the best nutritional sources of protein for humans and is consumed worldwide due to its highly appreciated taste [1]. A recent report issued by the Organization for Economic Cooperation and Development and Food and Agriculture Organization (OECD-FAO) revealed that the average annual global meat consumption surpassed 327 million tons (carcass weight equivalent) from 2018 to 2020 [2]. Due to differences in prices, unethical producers sometimes blend expensive meat with lower priced meat, such as by supplementing beef with pork to increase profits [3]. An economic loss of USD 45.6 million occurred in Europe due to beef products being adulterated with horse meat [4]. Besides the economic loss, the illegal activity of fraudulent substitution also raises serious concerns about food safety, public health, religion, and ethics [5]. Therefore, it is important to develop a reliable method for the detection of adulterated meat.

To date, the technologies that have been used in the detection of meat adulteration include biology-based, chemistry-based, and spectroscopy-based methods. Biology-based technologies have included polymerase chain reaction (PCR) [6], polymerase chain reaction-restriction fragment length polymorphism (PCR-RFLP) [7], loop-mediated isothermal amplification (LAMP) [8], and enzyme-linked immunosorbent assay (ELISA) [9]. Although these methods have proven to be reliable, specific, and sensitive, biology-based technologies are time-consuming, expensive, and require complex laboratory procedures to be performed by skilled personnel [10,11]. Chemistry-based technologies, such as gas chromatography mass spectrometer (GCMS) [12] and liquid chromatography-tandem mass spectrometer (LC-MS/MS) [13], have been proposed for halal verification and the accurate identification of meat products. Chemistry-based technologies are reliable and precise in the identification of adulterated meat. However, they require complex extractions and have long analysis times which significantly limit their widespread use [14]. For spectroscopy-based technologies, near-infrared spectroscopy (NIRS) [15], Raman spectroscopy (RS) [16], and hyperspectral imaging (HSI) [17] have been shown to be useful for detecting meat adulteration. However, the complex spectroscopy data requires a high degree of technical expertise to analyze. Due to the various drawbacks of these methods, it is imperative to develop a fast, precise, simple, and low-cost method for the detection of meat adulteration.

An electronic nose (E-nose) is a chemical measurement system used to measure the chemical properties of volatile gases and has been widely applied to detect the quality and safety of food due to its fast speed, high reliability, simple operation, and relatively low cost [18,19]. In recent years, reports of detecting meat adulteration using an E-nose have been increasing. Tian et al. [14] built a backpropagation neural network (BPNN) model for the prediction of pork content in minced mutton using a metal oxide semiconductor (MOS)-based E-nose and obtained a root mean square error (RMSE) of 5.26% on a test set. Han et al. [20] proposed a BPNN model for detecting pork adulteration in beef using a low-cost E-nose based on colorimetric sensors and got an RMSE of 0.147. Sarno et al. [21] proposed an optimized E-nose system using an optimized support vector machine (SVM) for the identification of pork in beef products and got an accuracy of 98.10%. To the best of our knowledge, most studies that have utilized E-nose devices have achieved good results in qualitative detection of meat adulteration; however, few studies have demonstrated reliable quantitative detection. An important reason underlying the imperfect performance in quantitative detection is that these studies have depended on manually selecting and extracting features. These manual operations not only burden the user with complex tasks but are also likely to lose valuable information.

The present work therefore aimed to propose a precise method for the quantitative detection of minced beef adulterated with pork. A high-efficiency framework (1DCNN-RFR) consisting of a 1DCNN backbone and an RFR is proposed for the quantitative detection of meat adulteration. The 1DCNN backbone is a powerful feature extractor that automatically mines the volatile compound information of meat samples. The RFR is employed as the regressor to strengthen the anti-overfitting ability, instead of the fully connected layer, for predicting adulterated proportions.

## 2. Materials and Methods

### 2.1. Meat Sample

In this study, fresh beef and pork satisfying Chinese national food safety standards [22] were purchased from a Carrefour supermarket in Beijing, China, and transported to the author’s laboratory in 20 min. Once in the laboratory, the fat and connective tissue were removed and the beef and pork were minced for 1 minute using a commercial blender (ZG-L805, Guangdong Zhigao Co. Ltd., Foshan, China). The minced beef was adulterated by mixing it with minced pork at seven distinct proportions by weight (0%, 10%, 20%, 30%, 40%, 50%, and 60%). Adulterated proportions higher than 60% were not considered because they are easily identifiable by human senses.

To undertake a more comprehensive evaluation of the models, two independent measurements (Measurement A and Measurement B) of sample gases were taken by the E-nose every day. For Measurement A, meat samples were purchased and measured in the morning. For Measurement B, meat samples were purchased and measured in the afternoon. Every day, three samples of each adulterated proportion were prepared, making 21 samples (3 samples of each proportion × 7 proportions) for both Measurement A and Measurement B. The daily operations for making experiment samples were same and lasted for 10 days. A total of 420 samples (10 days × 2 measurements × 21 samples) were measured. The details of the meat samples are shown in Table 1 and Table 2.

### 2.2. Data Collection by E-Nose

In this study, a PEN3 E-nose (Airsense Analytics GmbH, Schwerin, Germany) was used to collect the E-nose data of the adulterated meat. The PNE3 E-nose has a sensor array with 10 different MOS sensors. The details of the sensors are listed in Table 3.

All experiments were conducted in a single clean laboratory room (about 45 square meters) with the temperature and relative humidity were controlled at 25 ± 1 °C, 50 ± 2%, respectively. Measurements of the sample gases were conducted in a well-ventilated location to reduce baseline fluctuations and interference from other gas molecules. The air in the working environment was filtered by two active charcoal filters (Filter 1 and Filter 2 in Figure 1) to produce the zero gas, which was used as the baseline in this study.

The workflow of E-nose data collection included a collection stage and cleaning stage, as shown in Figure 1. Before collecting E-nose data, the meat samples were placed in sealed 50 mL samplers for 3 min so that the volatile gases from the samples would fill the sampler airspace. During the collection stage, zero-point trim was first conducted via a 15 s automatic adjustment and calibration of the zero gas. The values relative to the zero-point values were recorded as a baseline. After the establishment of the baseline, the volatile gases from the meat samples were pumped into the sensor chamber following Arrow 1 in Figure 1 at a constant flow rate of 10 mL/s where they contacted the MOS sensors. In this manner, the gas molecules were absorbed on the sensors’ surfaces where they changed the sensors’ conductivity through redox reactions with the sensors’ active elements [23]. The sensors’ conductivities stabilized when they were saturated. The sample gases were measured every 1 s for 100 s during the collection stage. During the cleaning stage, clean air from Filter 2 was pumped into the sensor chamber in the direction of Arrow 2 in Figure 1 to entirely remove the substances that had absorbed on the surface of the sensors. The cleaning stage lasted 30 s. As a result, the measurement of each sample lasted 325 s. Figure 2 shows examples of response curves of the 10 sensors’ during the collection stage for the 7 different proportions of adulterated meat. Each response curve represents the ratio of G (the conductivity of the sensor when contacted by the sample’s volatile gases) to G0 (the conductivity of the sensor when contacted by the zero gas). As shown in Figure 2, the response curves rose slowly and steadily from 0 s to 90 s and stabilized after 90 s.

### 2.3. Dataset Settings

The stable value (SV) is a vital and simple feature of the E-nose response signal which reflects the properties of the substances in the volatile gas and can be used by pattern recognition algorithms [24]. In this study, SVs (the values from 91 s to 100 s of the response signal) were used as the input data to the SVR, RFR, and BPNN. The response values from whole data collection stage (the values from 1 s to 100 s of the response signal) were used as the input data of the 1DCNN framework and the 1DCNN-RFR framework. The details of the datasets used in this study were as follows.

Dataset A: This dataset comprised a total of 210 samples (10 days × 3 samples for each proportion × 7 proportions) of Measurement A. For the SVR, RFR, and BPNN, the dataset could be expressed as a 2100 × 10 (number of sensors) matrix. For the 1DCNN and 1DCNN-RFR frameworks, the dataset could be expressed as a 210 × 10 × 100 matrix.

Dataset B: This dataset comprised a total of 210 samples (10 days × 3 samples for each proportion × 7 proportions) of Measurement B. For the SVR, RFR, and BPNN, the dataset could be expressed as a 2100 × 10 (number of sensors) matrix. For the 1DCNN and 1DCNN-RFR frameworks, the dataset can be expressed as a 210 × 10 × 100 matrix.

### 2.4. Related Works

#### 2.4.1. Principal Component Analysis (PCA)

Principal component analysis (PCA) is a commonly utilized method for feature extraction [25]. PCA is mathematically defined as an orthogonal linear transformation that transforms data into a new coordinate system such that the greatest variance by some projection of the data comes to lie along the first coordinate (the first principal component), the second greatest variance along the second coordinate, and so on [26]. Generally, the first few principal components must contribute at least 85% of the variance, or else the PCA method would be considered unsuitable because too much of the original information would be lost [27]. The first few principal components to make up a cumulative contribution exceeding 95% contain nearly all the information of the original data [24]. PCA is arguably the most popular multivariate statistical technique and has been applied in nearly all scientific disciplines [28].

#### 2.4.2. Convolutional Neural Network (CNN)

The CNN is a supervised feed-forward deep learning network designed to process data that come in the form of multiple arrays [29]. Basically, CNN are composed of three types of layers: the convolutional, pooling, and fully-connected layers [30,31]. The convolutional layer is composed of several convolutional kernels which are used to compute different feature maps and the pooling layer merges semantically similar features into one and prevents overfitting. After several convolutional and pooling layers, there will be one or more fully connected layers which take all neurons from the previous layer and connect them to every single neuron of the current layer to generate global semantic information. CNNs have provided excellent performance solutions to various problems in image classification, object detection, games and decisions, and natural language processing [32].

#### 2.4.3. Random Forest (RF)

Random forest (RF) is a combination of tree predictors such that each tree depends on the values from a random vector sampled independently and where all the trees in the forest have the same distribution [33]. The aim of the RF is to create a large number of uncorrelated decision tree models to produce more accurate predictions [34]. According to the strong law of large numbers, an increasing number of decision tree models leads to better generalizations and prevention of overfitting [35]. For the construction of the RF, N bootstrap samples are first drawn from the original training set (with replacement). Then, for each bootstrap sample, an unpruned classification or regression tree is grown with the following modifications: at each node, random sample m (m < M) of the predictors (each sample contains M predictors) is taken and the best split from among those variables is chosen. The second step is repeated until the node can no longer be split without pruning. Finally, the generated decision trees are formed into a random forest that is used to classify or regress the new data [36,37]. Compared to other machine learning methods, RF has various advantages, including low complexity, fast computing speed, lower overfitting, etc. [38].

#### 2.4.4. Evaluation Metrics

In this study, three evaluation metrics, including the coefficient of determination ($R^{2}$), the root mean square error (RMSE), and the mean absolute error (MAE), were used to evaluate the regression performance of the four models and the proposed framework. The $R^{2}$ is usually presented as an estimate of the percentage of variance within the response variable explained by its (linear) relationship with the explanatory variables [39]. The RMSE represents the standard deviation of the differences between the predicted values and the observed values of the samples [19]. The MAE is defined as the average absolute difference between the predicted values and the observed values of the samples. The evaluation metrics are defined in Equations (1)–(3).
(1)$$R^{2}=1-\frac{\sum_{i=1}^{n}{\left(y_{i}-{\hat{y}}_{i}\right)}^{2}}{\sum_{i=1}^{n}{\left(y_{i}-\bar{y}\right)}^{2}}$$
(2)$$RMSE=\sqrt{\frac{\sum_{i=1}^{n}{\left(y_{i}-{\hat{y}}_{i}\right)}^{2}}{n}}$$
(3)$$MAE=\frac{1}{n}\sum_{i=1}^{n}\left|y_{i}-{\hat{y}}_{i}\right|$$
where *n* is the number of the samples in the training set or the test set; $y_{i}$ is the actual value of the ith sample; ${\hat{y}}_{i}$ is the predicted value of the ith sample; and $\bar{y}$ is the mean of the actual value.

## 3. Proposed Method

Although the RFR has a low complexity and a lower likelihood of overfitting, its poor ability to extract a sufficient number of features may limit its performance. The 1DCNN is a type of CNN model [40] which only performs one-dimensional convolutions, giving them simpler structures and fewer parameters. Therefore, 1DCNN can save on computing resources and time [41]. Generally, a 1DCNN with the appropriate structure can mine sufficient features from the input data and with a flexible form. However, the 1DCNN requires more samples for training than traditional statistical models and the number of samples required may make it impractical for E-nose applications, where it may result in overfitting. Thus, a 1DCNN-RFR framework (presented in Figure 3a) consisting of a 1DCNN backbone (presented in Figure 3b) and an RFR was developed to combine the advantages of the 1DCNN and RFR to predict the proportions of pork in adulterated minced beef.

### 3.1. Data Preprocessing

In recent years, data preprocessing has increased in importance because data-mining algorithms require meaningful and manageable data so that they can correctly operate and provide useful knowledge, predictions, and descriptions. As Table 3 shows, the 10 MOS sensors exhibited strong cross-sensitivity because the sensitive substances of the sensors had some overlap. This is why E-nose data needs to be preprocessed before determining correlations between the sensors for the input of the 1DCNN backbone. Thus, the data preprocessing step is crucial to improving the prediction precision. To fit the multichannel input used in the 1DCNN model, the raw E-nose data were converted into a 10-channel input for the 1DCNN backbone. As shown in Figure 3a, the raw E-nose data formed a matrix with a size of 100 × 10, where 100 represents the 100 data points in the data collection stage (100 s) from each sensor and 10 represents the 10 MOS sensors. The 100 raw data points from each sensor were converted into a vectors with lengths of 100, and the 10 vectors of the sensors were concatenated into a 10 × 100 matrix as a 10-channel input of the 1DCNN backbone.

### 3.2. Proposed 1DCNN-RFR Framework

In this study, a 1DCNN backbone consisting of four convolutional layers and three pooling layers was constructed to automatically extract features from the E-nose data. For all convolutional operations, the convolutional kernel size was 3. A batch normalization layer was used to accelerate convergence and improve the generalization ability of the operation. ReLU was selected as the activation function to increase the ability of nonlinear expression. In all downsampling operations, the stride was 2 and the filter size was 2 for the first two pooling layers (Pool 1 and Pool 2). For the last pooling layer (Pool 3), the stride was 1 and the filter size was 3. A flattening layer came last, wherein a flattening transformation is applied to the tensor to convert the two-dimensional matrix of features into a vector that can be fed into the RFR. The specific design of the layers is shown in Table 4. The RFR, which consists of 100 random decision trees, is used to predict the adulterated proportions. Each tree is grown based on a resampling, using a regression tree procedure with a random subset of variables selected at each node.

As shown in Figure 4, the 1DCNN-RFR framework workflow begins with the 1DCNN backbone automatically extracting features from the E-nose data, which are then fed to the RFR to output the adulterated proportions. An MSE loss function serves to calculate error and is minimized by the stochastic gradient descent (SGD) optimizer. A learning decay strategy with a fixed step size is used during training of the 1DCNN backbone. The batch size and the initial learning rate were set to 49 and 0.001, respectively. After the training process of the framework is completed (solid line), the test samples are used to verify the performance of the trained framework (dotted line).

## 4. Results and Discussion

### 4.1. Principal Component Analysis

In pattern recognition, principal component analysis (PCA) is commonly used for feature extraction. In this study, the PCA was used to intuitively show the data distribution in a low-dimensional space by extracting meaningful information from the raw E-nose data. The SVs of 63 randomly selected meat samples (9 samples of each proportion × 7 proportions) were used in the PCA. The first three principal components contributed 92.76% of the variance and were selected to visualize the PCA results. The projections of the first three principal components are shown in Figure 5. The three axes represent the first principal component (PC 1), the second principal component (PC 2), and the third principal component (PC 3), which contributed 58.53%, 26.40%, and 7.83%, respectively. As shown in Figure 5, the PCA was able to distinguish pure beef from adulterated meat. However, there was a large overlap between the adulterated meats of different proportions. Therefore, the PCA was unable to distinguish among meat samples with different proportions of adulteration.

### 4.2. Comparing Five Models

Support vector regression (SVR), as an extension of the support vector machines in regressions, is commonly applied in a variety of fields [42]. BPNN, as a known machine learning method, is proven to have great potential in rapid detection of meat adulteration [14,20]. Therefore, an SVR model with RBF kernel and a BPNN model were also examined in this study. The performances of the SVR model, RFR model, BPNN model, and 1DCNN framework were compared with that of the proposed 1DCNN-RFR framework using multiple evaluation metrics, i.e., $R^{2}$, RMSE, and MAE. The 1DCNN framework consisted of the 1DCNN backbone and a 1DCNN regressor. The 1DCNN regressor consisted of two fully-connected layers with 64 and 32 neurons, respectively. ReLU and Sigmoid were selected as the activation functions for the first and second fully connected layers to strengthen the nonlinear expression ability of the 1DCNN regressor. A grid search method was used for identifying the best parameters of the five models via a three-fold cross-validation using the training set. The five models were programmed using the scikit learn library and the open-source PyTorch framework. Two experiments (Experiment A and Experiment B) with two datasets (Dataset A and Dataset B) were conducted to thoroughly compare the performances of the five models.

#### 4.2.1. Experiment A

Dataset A was used in Experiment A, and was divided into two parts, namely the training set (data from the first 7 days, 147 samples) and the test set (data from the remaining 3 days, 63 samples). For the SVR, RFR, and BPNN models, the training set was expressed as a 1470 × 10 matrix, the test set was expressed as a 630 × 10 matrix. For the 1DCNN and 1DCNN-RFR frameworks, the training set and the test set were obtained from all the response values and were expressed as a 147 × 10 × 100 matrix and a 63 × 10 × 100 matrix, respectively.

The SVR, RFR, and BPNN models, combined with SVs, were used to study the quantitative detection of adulteration in meat. The parameters searched for by the SVR model were the penalty factor (1, 5, 10, 20, 50, 100, 200, 500) and gamma (0.01, 0.1, 1, 5, 10, 20). The parameters searched for by the RFR model were the max depth (3, 5, 7, 9, 11, 13) and the minimum number of samples required to split an internal node (min. samples split: 7, 14, 21, 28, 35, 42). Through the grid search skill, the penalty factor and gamma for the SVR model were set to 50 and 1, respectively. For the RFR model, the max depth and the min samples split were set to 5 and 35, respectively. For the BPNN model, the optimum network architecture was obtained with topological architecture 10-21-1. An MSE loss function serves to calculate error and is minimized by the stochastic gradient descent (SGD) optimizer. ReLU was selected as the activation function of the BPNN model. The regression results from the SVR, RFR, and BPNN models on the test set are shown in Table 5. Three evaluation metrics, the $R^{2}$, RMSE, and MAE, were computed to comprehensively and accurately assess these two regression models. When combined with SVs, the BPNN model achieved a marginally better result with an $R^{2}$ of 0.9456, an RMSE of 4.6663%, and a MAE of 2.6486%. However, the performances of these three models were unsatisfactory because they failed to extract a sufficient number of features.

The 10-channel input was submitted to the proposed 1DCNN framework consisting of the 1DCNN backbone and a 1DCNN regressor. The 1DCNN framework was implemented on a laptop (Intel Core i7-9750H processor). The number of epochs was 1100 and the training time was 57.6207 s. The training loss of the 1DCNN framework during the training process is shown in Figure 6. The 1DCNN framework converged after 920 epochs. The parameters at 920 epochs were saved and used to evaluate the performance of the 1DCNN framework on the test set. As shown in Table 5, the 1DCNN framework performed much better than the SVR and RFR models, with an $R^{2}$ of 0.9852, an RMSE of 2.4301%, and a MAE of 2.0250% on the test set. The comparison with the SVR, RFR, and BPNN models, it was revealed that automatically mining a sufficient number of features from the E-nose data using the 1DCNN backbone significantly improved detection performance.

The 10-channel input was submitted to the proposed 1DCNN-RFR framework consisting of the 1DCNN backbone and the RFR. The 1DCNN backbone in the 1DCNN-RFR framework used the model parameters of the trained 1DCNN framework at 920 epochs. The training stage of the RFR in the 1DCNN-RFR framework was the same as that of the RFR model. Through the grid search skill, the max. depth and the min. samples split were set to 5 and 35, respectively. As shown in Table 5, the proposed 1DCNN-RFR framework achieved a better performance than all other models, with an $R^{2}$ of 0.9977, an RMSE of 0.9491%, and a MAE of 0.4619% on the test set. These results indicated that the strong prediction ability of the RFR improved the regression performance in the 1DCNN-RFR framework. The relationships between the predicted adulterated proportions by the five models and the corresponding actual adulterated proportions are shown in Figure 7. The x-axis represents the sequence number of the tested samples, and the black and red points along the y-axis represent the actual and predicted proportions, respectively. Figure 7 intuitively illustrates that the predictive performances of models using the 1DCNN backbone to extract features were significantly better than those that did not use the 1DCNN backbone. The proposed 1DCNN-RFR framework achieved the best predictions and predicted almost all adulterated proportions precisely.

#### 4.2.2. Experiment B

In practical applications, the number of samples will probably be much more limited than in Experiment A. Thus, Experiment B used a smaller number of training samples to further evaluate the generalization performance of the models. Dataset B was used in this experiment and was divided into the training set (data from the first 3 days, 63 samples) and the test set (data from the remaining 7 days, 147 samples). For the SVR, RFR, and BPNN models, the training set was expressed as a 630 × 10 matrix and the test set was expressed as a 1470 × 10 matrix. For the 1DCNN and 1DCNN-RFR frameworks, the training set and the test set were expressed as a 63 × 10 × 100 matrix and a 147 × 10 × 100 matrix, respectively.

All the experimental steps were the same as those of Experiment A, except for the 1DCNN where the batch size was adjusted from 49 to 21, while the other parameters were left unchanged. The best parameters of the SVR and RFR models were a penalty factor of 500, gamma of 0.1, max. depth of 3, and min. samples split of 21. The max. depth and min. samples split of the RFR in the 1DCNN-RFR framework were set to 3 and 7, respectively. The training loss of the 1DCNN during training is shown in Figure 8. The 1DCNN framework converged after 980 epochs. The parameters at 980 epochs were saved and used to evaluate the performance of the 1DCNN framework on the test set. The test set regression results from the five models (the SVR model, RFR model, BPNN model, 1DCNN framework, and 1DCNN-RFR framework) are shown in Table 6. The prediction performances of the SVR and RFR models in Experiment B were much worse than those of Experiment A. The BPNN model, the 1DCNN framework, and the proposed 1DCNN-RFR framework also suffered a slight reduction in performance. Even so, the 1DCNN framework and the 1DCNN-RFR framework performed much better than the SVR and RFR models. The 1DCNN-RFR model still worked best and obtained a good result with an $R^{2}$ of 0.9858, an RMSE of 2.3849%, and a MAE of 1.1625% on the test set. The regression results in Experiment B further demonstrated the superiority of the proposed 1DCNN-RFR framework, which performed well despite the much smaller sample size. The relationships between the predicted adulterated proportions of the five models and the actual adulterated proportions are shown in Figure 9. These relationships showed that the SVR and RFR models, which did not use the 1DCNN backbone and were unable to extract a sufficient number of features, had extremely poor prediction results. The proposed 1DCNN-RFR framework performed best.

## 5. Conclusions

In this study, a novel framework 1DCNN-RFR, consisting of a 1DCNN backbone and an RFR, was proposed for the quantitative detection of beef adulterated with pork using an MOS-based E-nose. The 1DCNN backbone automatically extracted a sufficient number of features from the E-nose data. The RFR strengthened the generalization ability of the 1DCNN framework and improved the prediction performance. Compared with the other four models (SVR, RFR, BPNN, and 1DCNN), the proposed 1DCNN-RFR framework obtained the best results on the test set, with an $R^{2}$ of 0.9977, an RMSE of 0.9491%, and a MSE of 0.4619%. All the findings suggest that the MOS-based E-nose coupled with the proposed 1DCNN-RFR framework has great potential for the quantitative detection of minced beef adulterated with pork.

## Figures and Tables

**Figure 1 foods-11-00602-f001:**
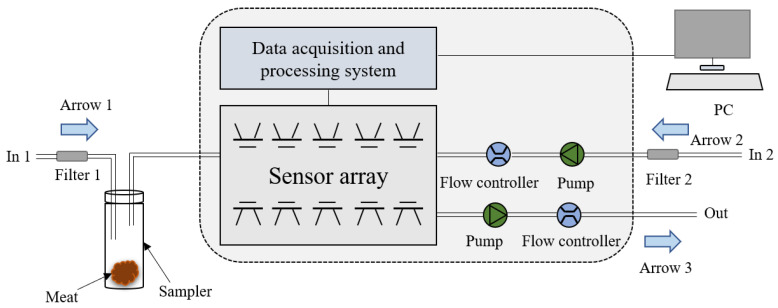
Schematic diagram of data collection using an E-nose.

**Figure 2 foods-11-00602-f002:**
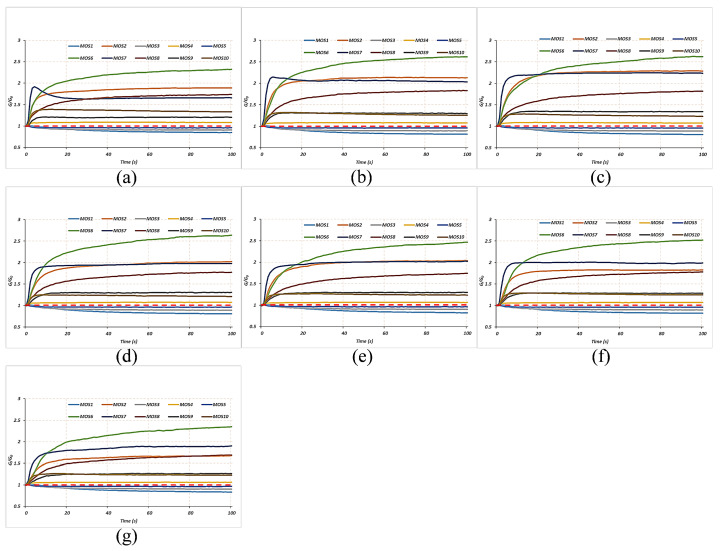
Example response curves of the 10 E-nose sensors (the baseline is marked with the red dotted line): (**a**) proportion of 0%; (**b**) proportion of 10%; (**c**) proportion of 20%; (**d**) proportion of 30%; (**e**) proportion of 40%; (**f**) proportion of 50%; (**g**) proportion of 60%.

**Figure 3 foods-11-00602-f003:**
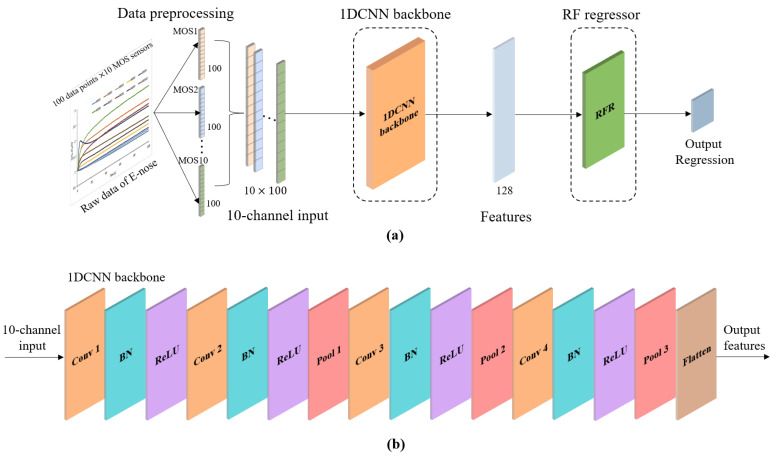
Schematic diagram of the proposed framework. (**a**) Structure of the proposed 1DCNN-RFR framework; (**b**) structure of the proposed 1DCNN backbone.

**Figure 4 foods-11-00602-f004:**
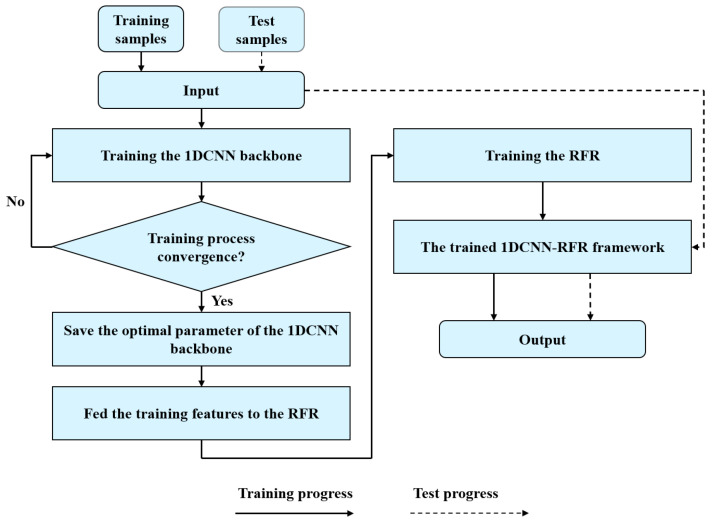
The workflow of the 1DCNN-RFR framework.

**Figure 5 foods-11-00602-f005:**
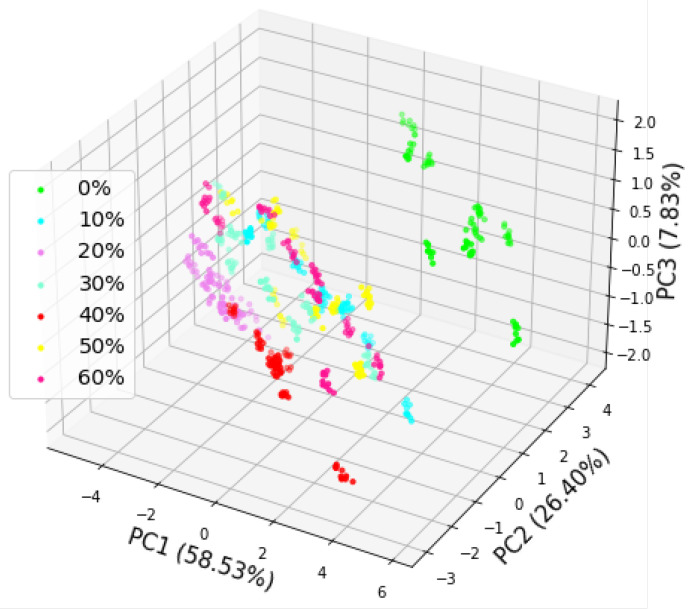
Projection of the first three principal components of the PCA of the meat samples.

**Figure 6 foods-11-00602-f006:**
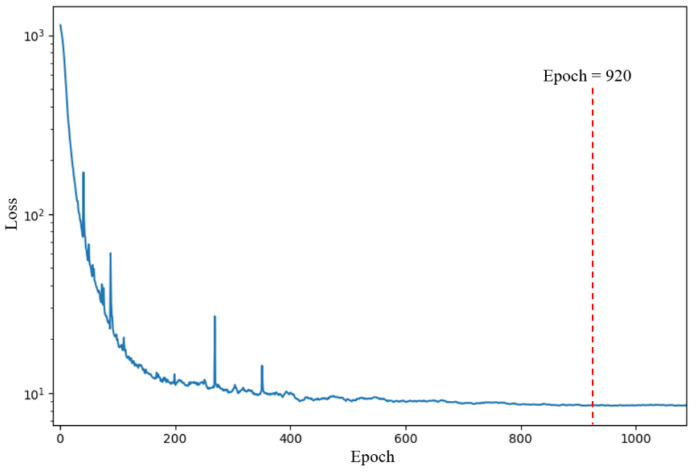
Training loss curve of the 1DCNN framework during training in Experiment A.

**Figure 7 foods-11-00602-f007:**
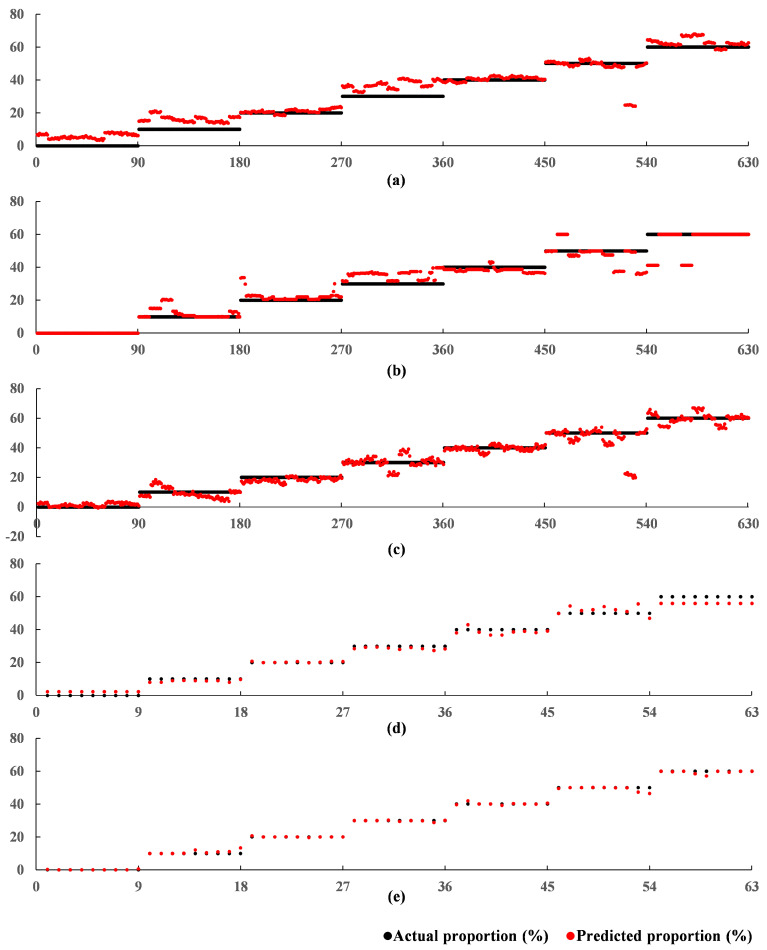
Relationships between the predicted adulterated proportions of the five models and the actual adulterated proportions in Experiment A: (**a**) SVR; (**b**) RFR; (**c**) BPNN; (**d**) 1DCNN; (**e**) 1DCNN-RFR.

**Figure 8 foods-11-00602-f008:**
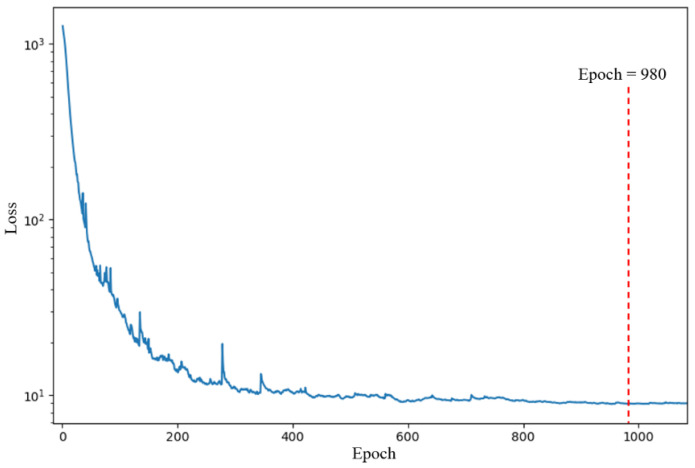
Training loss curve of the 1DCNN framework during training in Experiment B.

**Figure 9 foods-11-00602-f009:**
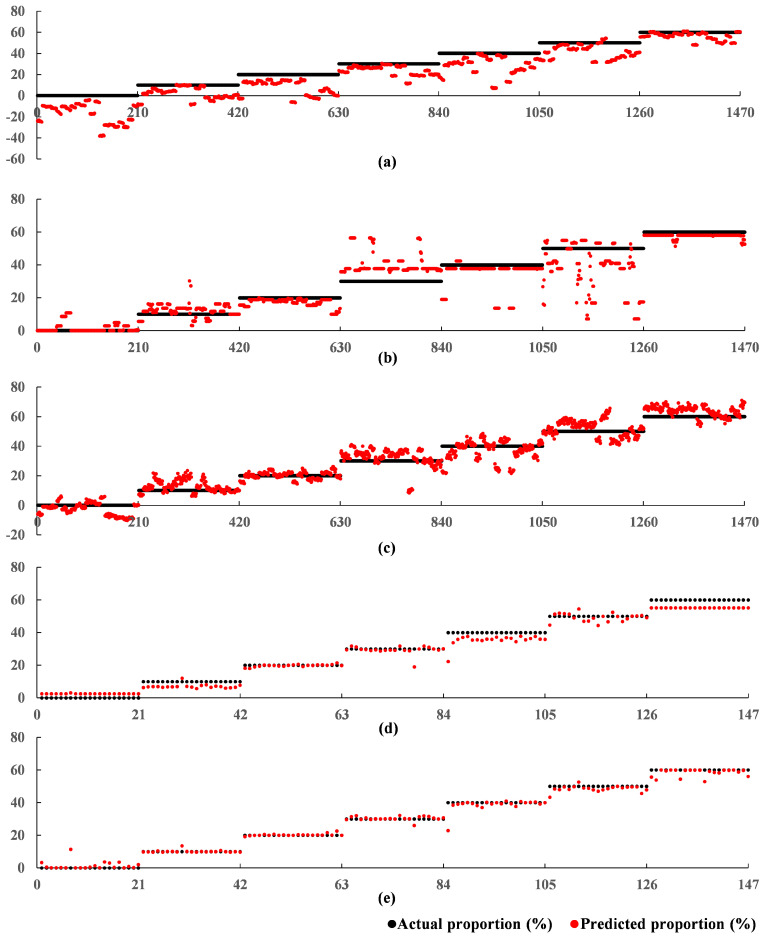
Relationships between the predicted adulterated proportions of the five models and the actual adulterated proportions in Experiment B. (**a**) SVR; (**b**) RFR; (**c**) BPNN; (**d**) 1DCNN; (**e**) 1DCNN-RFR.

**Table 1 foods-11-00602-t001:** Details of the meat samples in Measurement A.

No.	Adulterated Proportion	Measuring Time	Number of Samples
Sa.P1	0%	Morning	30 (10 days × 3 samples)
Sa.P2	10%	Morning	30 (10 days × 3 samples)
Sa.P3	20%	Morning	30 (10 days × 3 samples)
Sa.P4	30%	Morning	30 (10 days × 3 samples)
Sa.P5	40%	Morning	30 (10 days × 3 samples)
Sa.P6	50%	Morning	30 (10 days × 3 samples)
Sa.P7	60%	Morning	30 (10 days × 3 samples)

**Table 2 foods-11-00602-t002:** Details of the meat samples in Measurement B.

No.	Adulterated Proportion	Measuring Time	Number of Samples
Sa.P1	0%	Afternoon	30 (10 days × 3 samples)
Sa.P2	10%	Afternoon	30 (10 days × 3 samples)
Sa.P3	20%	Afternoon	30 (10 days × 3 samples)
Sa.P4	30%	Afternoon	30 (10 days × 3 samples)
Sa.P5	40%	Afternoon	30 (10 days × 3 samples)
Sa.P6	50%	Afternoon	30 (10 days × 3 samples)
Sa.P7	60%	Afternoon	30 (10 days × 3 samples)

**Table 3 foods-11-00602-t003:** Sensor array details [23].

No.	Sensor	Main Performance
1	W1C	Sensitive to aromatic compounds
2	W5S	High sensitivity to nitrogen oxides, broad range sensitivity
3	W3C	Sensitive to ammonia and aromatic compounds
4	W6S	Sensitive mainly to hydrogen
5	W5C	Sensitive to alkanes and aromatic components and less sensitive to polar compounds
6	W1S	Sensitive to methane, broad range sensitivity
7	W1W	Sensitive primarily to sulfur compounds and many terpenes and organic sulfur compounds
8	W2S	Sensitive to ethanol and less sensitive to aromatic compounds
9	W2W	Sensitive to aromatic compounds and organic sulfur compounds
10	W3S	Highly sensitive to alkanes

**Table 4 foods-11-00602-t004:** Details of the 1DCNN backbone.

No.	Layer	Parameters	Outpur Size
Input	Input	/	10 × 100
Conv 1	ConvBNReLU	16@3, padding = 1, stride = 2	16 × 50
Conv 2	ConvBNReLU	32@3, padding = 0, stride = 1	32 × 48
Pool 1	Max Pooling	Filter size = 2, stride = 2	32 × 24
Conv 3	ConvBNReLU	64@3, padding = 1, stride = 2	64 × 12
Pool 2	Max Pooling	Filter size = 2, stride = 2	64 × 6
Conv 4	ConvBNReLU	128@3, padding = 1, stride = 2	128 × 3
Pool 3	Avg Pooling	Filter size = 3, stride = 1	128 × 1
Flatten	Flatten	/	128

**Table 5 foods-11-00602-t005:** Evaluation of the regressions from the five models using the test set in Experiment A using three metrics.

Model	$R^{2}$	RMSE (%)	MAE (%)
SVR	0.9183	5.7176	4.1394
RFR	0.9290	5.3293	2.9782
BPNN	0.9456	4.6663	2.6486
1DCNN	0.9852	2.4301	2.0250
1DCNN-RFR	0.9977	0.9491	0.4619

**Table 6 foods-11-00602-t006:** Evaluation of regressions on the test set by the five models in Experiment B using three metrics.

Model	$R^{2}$	RMSE (%)	MAE (%)
SVR	0.6108	12.4770	9.7819
RFR	0.7734	9.5204	5.6891
BPNN	0.9219	5.5910	4.2554
1DCNN	0.9703	3.4441	2.6853
1DCNN-RFR	0.9858	2.3849	1.1625

## Data Availability

The data presented in this study are available at Figshare (https://doi.org/10.6084/m9.figshare.19200284.v1).

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
