# Peer review of "A Machine Learning Method for the Quantitative Detection of Adulterated Meat Using a MOS-Based E-Nose"

_foods, 2022, doi:10.3390/foods11040602_

Round 1

Reviewer 1 Report

The article titled "A Machine Learning Method for the Quantitative Detection of Adulterated Meat Using a MOS-Based E-Nose" deals with Machine learning methods applied to a sensor array. The article deals with a very complex problem that is difficult to solve effectively with the analysis of VOCs alone. I believe that the data treated here are very poor but translated with advanced machine learning methods we try to make sense.
Some notes that might be helpful:
- I recommend doing even lower%
- do you measure the same sample for 10 days? so is it a shelf-life analysis? of meat that is no longer edible? have you considered doing microbiological analyzes of the sample?
- how were the 10 sensors chosen? based on what characteristics? what compounds do you think can be generated after 10 days of analyzing a sample (biogenic amines?)
- how long does the measurement last in total? tiga 115 to 127
- at what T ° are the samples kept during sampling?
- figure 2 does not make sense, ask to add: A) the same sensor as it varies at different%. B) to make all the measurements between sampling and recovery including the base line. (which R / R0?)
- it seems that there are several sensors that do not give any useful contribution, that is they are not able to discriminate why they are used?

Reviewer 2 Report

Dear Authors,

A PDF file with review is attached.

Regards
